# Simulation on the Permeability Evaluation of a Hybrid Liner for the Prevention of Contaminant Diffusion in Soils Contaminated with Total Petroleum Hydrocarbon

**DOI:** 10.3390/ijerph192013710

**Published:** 2022-10-21

**Authors:** Jeongjun Park, Gigwon Hong

**Affiliations:** 1Incheon Disaster Prevention Research Center, Incheon National University, 119 Academy-ro, Yeonsu-gu, Incheon 22012, Korea; 2Department of Civil Engineering, Halla University, 28 Halladae-gil, Wonju-si 26404, Korea

**Keywords:** hybrid liner, permeability, contaminant, diffusion

## Abstract

This study describes the test results to evaluate the impermeability efficiency, according to the total petroleum hydrocarbon (TPH) reaction time of a hybrid liner for preventing the TPH diffusion, and the numerical analysis results, according to the various TPH reaction times of the hybrid liner. The experimental results indicated that the hybrid liner performed effectively as an impermeable material under the condition of a 4 h reaction time between TPH and the hybrid liner. In other words, the permeability of the hybrid liner was lower than 7.64 × 10^−7^ cm/s when the reaction time of the TPH and the hybrid liner exceeded 4 h. This means that polynorbornene applied as a reactant becomes completely gelated four hours after it reacts with TPH, demonstrating its applicability as a liner. The numerical analysis results to evaluate the TPH diffusion, according to the hybrid liner-TPH reaction time indicated that the concentration decreased, compared to the initial concentration as the hybrid liner-TPH reaction time increased, regardless of the head-difference and the observation point for all concentration conditions. In addition, the reduction ratio of the concentration, compared to the initial concentration was 99% ~ 100%, when the reaction time of the hybrid liner-TPH was more than 4 h. It was found that the concentration diffusion of TPH reacting with the hybrid liner was decreased when the distance from the hybrid liner and the reaction time of the hybrid liner-TPH were increased. In other words, in the case of a high-TPH condition, the concentration reduction ratio is 12.5~17.8%, 16.9~29.7%, depending on the distance ratio (D/L = 0.06, 0.54, 0.94), respectively, when the reaction time of the hybrid liner-TPH is 0 h and 0.5 h, respectively. In the case of medium- and low-TPH conditions, the concentration reduction ratio, according to the distance ratio is 12.0% to 20.8% and 17.0% to 29.8%, respectively. This result means that a numerical analysis model can be used sufficiently to predict the TPH diffusion, according to the distance from the location where the hybrid liner is installed.

## 1. Introduction

As oil consumption is substantially increasing in industrial development, the environmental problem of oil pollution from oil storage tank facilities, such as industrial complexes and gas stations, is increasing [1,2]. A major contaminant of oil pollution is total petroleum hydrocarbon (TPH); when the soil becomes contaminated with TPH, the scale of the damage is enormous. Ławniczak et al. [3] reported that the contaminants with the largest impact on the environmental pollution are generated from crude oil-based hydrocarbons.

Traditionally, contaminated soil is rectified via ex situ or in situ remediation [4]. However, as it takes a long period of time to confirm the occurrence of contaminated soil, the contaminant is widely diffused throughout the contaminated soil. Particularly, for TPH pollution, remediating the contaminated soil is difficult and requires extensive time and costs [5]. As such, technologies must be developed to prevent the diffusion of the contaminants beforehand, such as liners installed in landfills in fields where the contaminants, such as TPH, can leak.

Several studies have been conducted to remediate the oil-contaminated soil. For example, researchers assessed the efficiency of contaminant removal via land farming and high-temperature thermal desorption on soil contaminated with petroleum hydrocarbon [6]. The contaminant removal mechanism of microwave heating in the soil contaminated with TPH has also been investigated [7]. Researchers have also studied the contaminant removal efficiency of the bioremediation and biopiles to remediate crude oil and TPH contaminants [8,9,10].

Numerous studies have also been conducted on the remediation technologies for oil-contaminated soil, using soil washing. One study examined a soil washing surface treatment method that desorbs the diesel contaminants from soil particles, using chemical oxidation and an aqueous solution containing a cleaning agent [11]. Another study evaluated the TPH removal efficiency from sand contaminated with diesel over a long period of time, using soil washing [12]. Soil washing and soil-flushing-based surfactants for remediating oil-contaminated soil, have been reported to have a high efficiency for the removal of diesel and TPH from the soil particles [13,14,15].

Various studies have also been performed on the effectiveness of advanced remediation techniques for oil-contaminated soil. For example, a study investigated the technical feasibility of remediating soil contaminated with biochar/graphite carbon nitride (BC/g-C_3_N_4_), to develop an eco-friendly remediation method [16]. Moreover, various studies related to the bioremediation of contaminants from crude oil, TPH, and petroleum-related products have been reviewed [8]. Researchers have studied the removal mechanism of TPH from soil via microwave heating [7], and experimentally evaluated the effectiveness of various remediation techniques (electroremediation, phyto-electrochemical, and electrooxidation) for areas contaminated with organic matter, based on electric fields [17].

The landfill liner system is a representative technique to prevent the diffusion of contaminants. In particular, geosynthetic clay liners (GCLs) are primarily applied to the liner system to prevent the diffusion of fluid-type contaminants, and related studies on this are being conducted. GCLs have low permeability coefficients and possess a mechanical stability. Moreover, they can stabilize the slope in addition to preventing the diffusion of contaminants, according to the reactive material, and thus they have diverse applications in geoenvironmental and geotechnical engineering fields [18,19,20,21].

As mentioned earlier, ex situ or in situ remediation is typically applied to remove oil contaminants in soil. However, when oil contaminants are identified on the surface, then, to prevent oil contaminant diffusion, incineration and recovery via the absorption of pollutants using oil-absorbing materials are generally applied [22]. Representative oil-absorbing materials include fabric-based oil absorbents, which are nonwoven fabrics made of hydrophobic hydrocarbon-based fibers [23,24]. To prevent oil contaminant diffusion more efficiently, research is being conducted on oil-absorbing resins, based on polymeric materials capable of gelation [25,26,27,28,29]. Many studies have been conducted on the repair of structures, based on various materials with absorption and expansion properties [30,31].

Based on a review of the various studies, we determined that most research related to the remediation of oil-contaminated soil focused on the contaminant removal and its efficiency. There were relatively few studies on preventing oil contaminant diffusion. In other words, most of the research focused on developing technologies for after an oil contaminant spill has occurred. This is because oil contaminant diffusion is caused by adsorption onto soil particles or the hydraulic properties of groundwater, which are difficult to physically control. Thus, a prior study developed and assessed the applicability of a reactive liner (named “geotextile-polynorbornene liner” in the previous study) that can respond to the hydraulic properties of groundwater and prevent oil contaminant diffusion via gelation [32].

In order to produce the developed reactive liner and apply it to the field, the experimental and analytical studies under various contaminant diffusion and long-term conditions must be continued. In general, the permeability of impermeable materials is evaluated by the methods presented in ASTM D5887 and ASTM D6766. The permeability of the hybrid liner applied in this study, was confirmed using the test method in a previous study [32]. However, it is necessary to evaluate the permeability of the hybrid liner according to the TPH diffusion in the test apparatus that can simulate the ground, because the previous study is not a test for soil. In other words, if the permeability of the developed hybrid liner is confirmed to be 1.0 × 10^−7^ cm/s through the experimental results, it can be evaluated as an impermeable liner. Therefore, in this study, a reactive liner with hydraulic properties and the effect of preventing TPH contaminant diffusion was developed. This liner was referred to as the “hybrid liner”. A test was conducted to evaluate its impermeability efficiency, according to the TPH reaction time of the hybrid liner. We quantitatively analyzed the permeability of the hybrid liner, based on the results of the impermeability efficiency evaluation test. Furthermore, compared to the previous study [32], the impermeability efficiency was assessed through simulations using various TPH reaction times of the hybrid liner.

## 2. Materials and Methods

### 2.1. Hybrid Liner

The hybrid liner uses the concept of the geosynthetic clay liner (GCL), which can absorb and swell the contaminants to achieve an impermeability efficiency. The concept is shown in Figure 1a. The hybrid liner consists of a reactive material and geosynthetics (Figure 1b). The reactive material (polynorbornene) that reacts to TPH, is restrained with needle-punched geotextiles on the upper and lower surfaces. The reaction principle of the hybrid liner is that before TPH leaks in the soil, groundwater has a normal flow because the groundwater does not react with polynorbornene. However, once TPH leaks and diffuses, it comes into contact with polynorbornene, causing absorption, expansion, and gelation of polynorbornene, as shown in Figure 2 [32]. Due to the reaction over time, an impermeable layer is formed because of the impermeability property of the hybrid liner, preventing further diffusion of TPH. The concept and principle of the hybrid liner are explained in detail in a previous study [32].

The maximum size of the hybrid liner is 3 m in width and 100 m in length. The thickness is 0.03 m, but the thickness can be increased when a large amount of reactive material is required, depending on the contamination level of TPH in the ground.

### 2.2. Experiment

The impermeability efficiency of the hybrid liner containing gelated polynorbornene can be assessed through a permeability test. The permeability coefficient of a typical impermeable layer is 10^−7^ cm/s or less. In tests for measuring the permeability of geosynthetics, such as geotextiles, a horizontal apparatus that maintains a constant water level is used to perform the evaluation. Accordingly, this study conducted a permeability test on a hybrid liner that reacted with TPH over time, based on a constant head-difference condition. Diesel oil was used as the source of TPH.

Figure 3 presents a schematic of the permeability test apparatus. The apparatus consists of an inlet and outlet tank of water and a separated circular pipe between the bottom of the inlet and outlet tanks, which can be used to simulate soil. The hybrid liner was installed and assembled in the center of the separated circular pipe, and all joints were processed and secured to avoid leakage of water and soils and to ensure the reliability of the permeability test results. The diameter of the hybrid liner installed in the permeability test apparatus for the lab. scale experiment is 0.1 m.

The procedure of the permeability test is summarized as follows. First, the separated left circular pipe was homogeneously filled with sand (Joomoonjin standard sand in South Korea). Sieve-type meshes, smaller than the sand particles, were installed on the left and right sides of the circular pipe, to prevent the sand particles from leaking and to enable a smooth water flow. Next, as the right circular pipe was connected, the hybrid liner, which was to react with TPH over time, was installed in the center, and a horizontal state was maintained. Water was then added to the inlet box to saturate the circular pipe, after which an outlet valve at a height that can maintain the water level was opened to release the water. Finally, the water flow was stabilized, based on the head-difference such that the water could uniformly flow out, after which the test was performed.

Table 1 lists the test cases. For each case, three tests were conducted under identical conditions to calculate the average permeability coefficient. Different specimen lengths were applied in each test case, due to the expansion of the hybrid liner, according to the TPH reaction time, whereas the specimen area of the hybrid liner installed in the center of the circular pipe was identical. Identical head conditions were also applied.

### 2.3. Numerical Analysis

In this study, a numerical analysis was conducted to perform the simulations according to various reaction times of the hybrid liner and TPH. For the numerical analysis, a finite difference analysis (FDA) was performed using Visual MODFLOW of the MT3D software. MT3D enables the convenient three-dimensional finite difference hydraulic model analysis for solute movement in complex hydrogeological structures. This software is widely used for contaminant diffusion analyses.

Figure 4 shows a 3D view and plan view of the FDA model. The TPH inlet box, for applying the hydraulic gradient, the soil with a permeability of 1.0 × 10^−4^ cm/s to simulate the soil conditions of the permeability test, the hybrid liner, and four observation points for measuring the TPH concentration of the soil passed through the hybrid liner, were modeled. Based on the total length of the soil (L = 0.5 m) behind the hybrid liner, two points were placed at 0.03 m (point 1, point 3), one point at 0.27 m (point 4), and one point at 0.47 m (point 2) from the hybrid liner. Table 2 lists the conditions for the FDA.

In general, a large variety of factors impact the movement of contaminants. However, in this FDA, the objective is to examine the soil concentration to verify that TPH has not leaked further after TPH leaks in the soil and reacts with the hybrid liner. Accordingly, a steady-flow state for the TPH movement was assumed in the FDA. In the previous study [32], the permeabilities for the TPH and hybrid liner reaction times of 0.5 h and 4 h, were obtained. By comparison, this study also analyzed the permeabilities for 0 h and 24 h. Using the prior results and additional analysis results, we evaluated the TPH diffusion rate, based on the maximum concentrations at the observation points, according to the reaction time of TPH and the hybrid liner. The results of the previous study were applied to the head condition and permeability coefficient of the hybrid liner. Table 3 shows the parameters for the FDA cases.

## 3. Results and Discussion

### 3.1. Permeability Evaluation of Hybrid Liner

Table 4 presents the permeability coefficients, based on the permeability tests on the hybrid liner, whereas Figure 5 presents a graph of the test results. For the hybrid liner that did not react with TPH (reaction time = 0 h), the average permeability coefficient was 1.11 × 10^−3^ cm/s, which is similar to the flow rate of groundwater in general sandy soil. This signifies that even if water comes into contact with the hybrid liner, the flow of the groundwater in the soil is maintained. Additionally, although these results slightly differ in terms of the permeability coefficients obtained via the permeability change tests conducted in the previous study [32], they can be regarded as similar.

At a hybrid liner-TPH reaction time of 0.5 h, the permeability coefficient decreased, but not to a level that could block fluid. By contrast, the permeability coefficients of the hybrid liner at hybrid liner-TPH reaction times of 4 h or more, rapidly decreased. The coefficient at 4 h, was 1.0 × 10^−7^ cm/s, which indicates an impermeable layer. Thus, the hybrid liner secured the performance of an impermeable material at a 4 h reaction time between TPH and the hybrid liner. Furthermore, when the reaction time was 48 h, the permeability coefficient of the hybrid liner was similar to that when the reaction time was 24 h. Hence, the polynorbornene applied as a reactant was completely gelated after 4 h, even when restrained by geosynthetics, demonstrating that it is sufficiently applicable as a reactant for the liner.

### 3.2. Evaluation of the TPH Diffusion, According to the Reaction Time of the Hybrid Liner and TPH

This study analyzed the results of the FDA that simulated the TPH flow reflecting the hybrid liner-TPH reaction time conditions. As mentioned in the previous study [32] and in Section 2.3, the results of the permeability change test of the hybrid liner that reacted with TPH over time, were applied to the permeability coefficient of the hybrid liner in the FDA. The TPH flow was generated for 96 h. In this section, the degree of diffusion is analyzed, based on the maximum concentration.

Table 5, Table 6 and Table 7 show the maximum concentration at each observation point for TPH in the soil that passed through the hybrid liner when the TPH flow was generated for 96 h, according to the TPH concentration condition. In other words, the values in Table 5, Table 6 and Table 7 mean that the results derived from the simulation (numerical analysis). Figure 6, Figure 7 and Figure 8 show the plots of the maximum concentrations at the observation points, according to the hybrid liner-TPH reaction time for each concentration condition. The reduction ratios of the concentration at the observation points for each analysis condition are also plotted.

Figure 6a shows a graph of the maximum concentrations listed in Table 5. For the high-TPH condition, the concentration decreased, compared to the initial concentration as the hybrid liner-TPH reaction time increased, regardless of the head-difference and the observation point. Moreover, the hybrid liner-TPH reaction time greatly decreased at all observation points from 4 h. These results were obtained using the concentration reduction ratio (Table 5) and presented as a graph in Figure 6b. Points 1 and 3, which are at the same distance from the hybrid liner, showed similar concentration reduction ratios. In addition, the concentration reduction ratio increased as the hybrid liner-TPH reaction time increased, even at the same head condition. When the hybrid liner-TPH reaction time was the same, the concentration reduction ratio tended to slightly increase as the head-difference increased. As with the observed maximum concentration, at hybrid liner–TPH reaction times of 4 h or more, the concentration reduction ratio compared to the initial concentration was 99% to 100%. This indicates that when the hybrid liner-TPH reaction time is 4 h or more, the hybrid liner can prevent the TPH diffusion by forming an impermeable layer that can block TPH.

Figure 7a shows a graph of the maximum concentrations in Table 6. For the medium-TPH condition, the concentration decreased, compared to the initial concentration as the hybrid liner-TPH reaction time increased, regardless of the head-difference and observation point. Moreover, the hybrid liner-TPH reaction time greatly decreased at all observation points at and beyond 4 h. These results are identical to the high-TPH condition results.

These results were calculated using the concentration reduction ratio (Table 6) and presented as a graph in Figure 7b. All trends were identical to those of the high-TPH condition results. Moreover, the calculated concentration reduction ratios, according to the hybrid liner-TPH reaction time, observation point, and head-difference were similar. The same trends were observed among the low-TPH condition results, shown in Table 6 and Figure 8. Hence, when the hybrid liner-TPH reaction time was 4 h or more, despite the slight differences in the concentration reduction ratio, the hybrid liner formed an impermeable state able to block TPH regardless of the concentration condition.

As described earlier, for each of the observation points, placed 0.03 m (point 1, point 3), 0.27 m (point 4), and 0.47 m (point 2), based on the total length of the soil (L = 0.5 m) behind the hybrid liner, the TPH concentration was observed, and the influence of the distance of the observation point was analyzed. If the observation points for the TPH concentration are expressed as distance ratios (D/L), point 1 (same as point 3), point 2, and point 4 are identified as 0.06, 0.94, and 0.54, respectively. Accordingly, the impermeable efficiency with respect to the distance ratio was analyzed using the concentration reduction ratio. For the impermeable efficiency, we considered an extreme situation limiting the FDA results by applying the head-difference condition with the lowest concentration reduction ratio. Figure 9 shows the results. 

With regard to the influence of the observation point distance, as the distance ratio from the hybrid liner was increased, the concentration reduction ratio increased regardless of the hybrid liner-TPH reaction time. The following is an evaluation of the observation point distance, excluding the conditions of a hybrid liner-TPH reaction time of 4 h or more, where the concentration reduction ratio is 99 to 100%. For the high-TPH condition, in which the hybrid liner-TPH reaction time was 0 h, the concentration reduction ratios were 12.5%, 22.1%, and 17.8%, with respect to distance ratios of D/L = 0.06, 0.54, and 0.94, respectively. When the hybrid liner-TPH reaction time was 0.5 h, the concentration reduction ratios were 16.9%, 24.0%, and 29.7% with respect to distance ratios of D/L = 0.06, 0.54, and 0.94, respectively. For the medium-TPH condition, in which the hybrid liner-TPH reaction times were 0 h and 0.5 h, the concentration reduction ratios were 12.0% to 20.8% and 17.0% to 29.8%, respectively, with respect to the distance ratio, which were the same as those for the low-TPH condition. Thus, the concentration diffusion of TPH that leaked in the soil and penetrated the hybrid liner decreased as the physical distance and hybrid liner-TPH reaction time increased.

In this study, limited materials and ground conditions were applied to simulate a lab. scale experiment using a hybrid liner. This has limitations in perfectly simulating the TPH diffusion in the field. Therefore, in order to secure the reliability of the simulation, various simulations that can simulate the field conditions (engineering properties and layer of soil) should be performed after performing field experiments applying the hybrid liner to the ground contaminated with TPH.

## 4. Conclusions

In this study, a test was conducted to evaluate the impermeability efficiency of a hybrid liner, which was designed to prevent the TPH contaminant diffusion, with respect to the TPH reaction time between the hybrid liner and TPH, and the permeability of the hybrid liner was quantitatively evaluated. Numerical analysis-based simulations were also performed, according to the various reaction times of TPH and the hybrid liner. The results are as follows.
(1)According to the results of the permeability tests on a hybrid liner that was made to react with TPH over time at a constant head-difference condition, the hybrid liner performed effectively as an impermeable material under the condition of the 4 h reaction time between TPH and the hybrid liner. These results show that the polynorbornene used as a reactant was completely gelated after 4 h, even when restrained by geosynthetics, demonstrating that it is sufficiently applicable as a liner material.(2)According to numerical analysis-based simulation results evaluating the TPH diffusion with respect to the hybrid liner-TPH reaction time, for all concentration conditions, the concentration decreased, compared to the initial concentration as the hybrid liner-TPH reaction time increased, regardless of the head-difference and the observation point. In particular, at hybrid liner-TPH reaction times of 4 h or more, the concentration reduction ratio, compared to the initial concentration, was 99% to 100%. This indicates that when the hybrid liner-TPH reaction time is 4 h or more, the hybrid liner is able to prevent the TPH diffusion by forming an impermeable layer that can block TPH. (3)According to an evaluation of the observation point distance, excluding the conditions of a hybrid liner-TPH reaction time of 4 h or more, where the concentration reduction ratio is 99 to 100%, the concentration diffusion of TPH that leaked in the soil and penetrated the hybrid liner decreased as the physical distance and the hybrid liner-TPH reaction time increased. This demonstrates that a numerical analysis model is sufficiently feasible for predicting the TPH diffusion, according to the distance from where the hybrid liner is installed.

## Figures and Tables

**Figure 1 ijerph-19-13710-f001:**
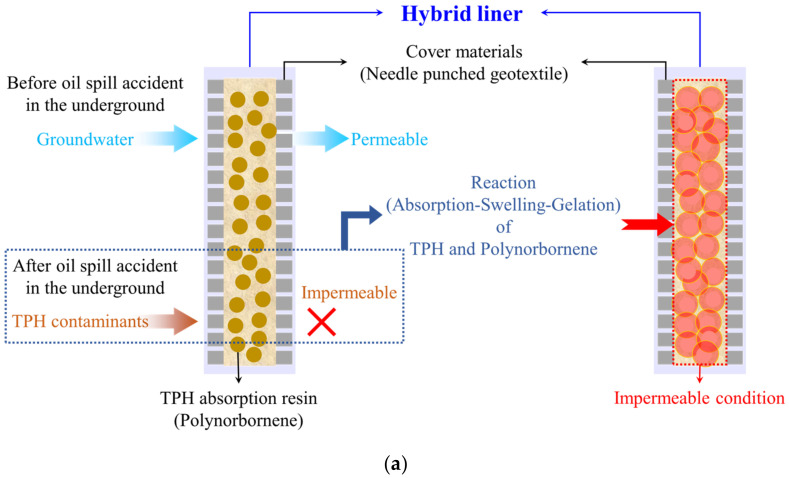
Hybrid liner: (**a**) Conceptualization; (**b**) Gelation over time (modified from [32]).

**Figure 2 ijerph-19-13710-f002:**
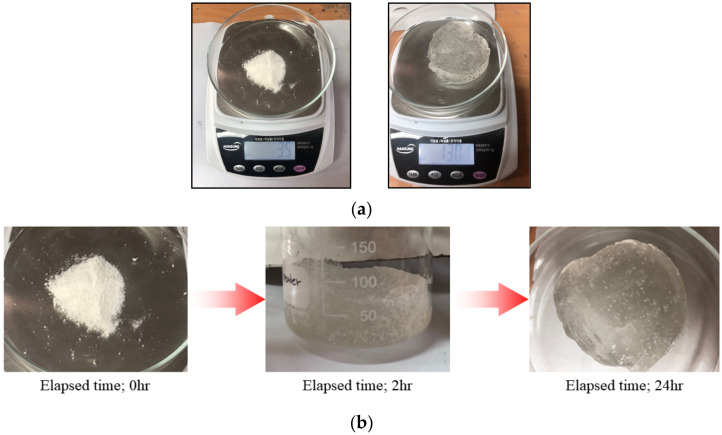
Absorption, expansion, and gelation of polynorbornene (reactive material): (**a**) Weight change after the reaction with TPH, before: 3.5 g, after: 130.1 g; (**b**) Gelation over time [32].

**Figure 3 ijerph-19-13710-f003:**
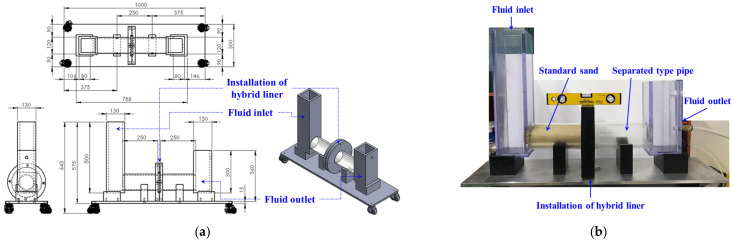
Permeability test apparatus: (**a**) Schematic; (**b**) Set-up.

**Figure 4 ijerph-19-13710-f004:**
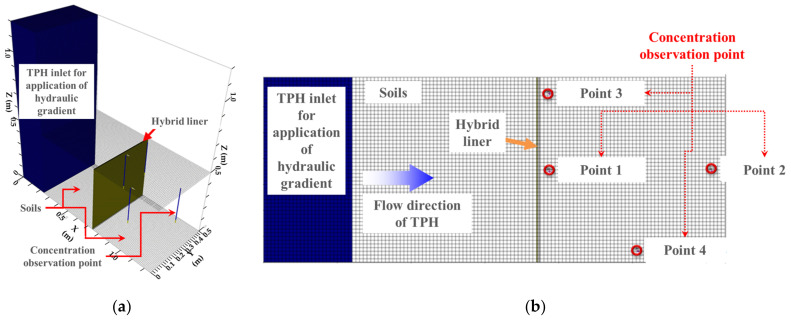
FDA model: (**a**) 3D view; (**b**) Plan view (modified from [32]).

**Figure 5 ijerph-19-13710-f005:**
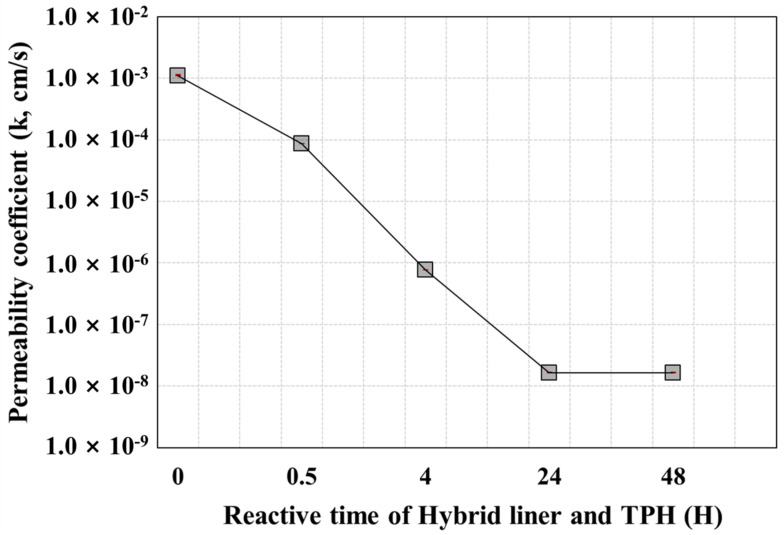
Variation of the permeability coefficient of the hybrid liner by reactive time of TPH.

**Figure 6 ijerph-19-13710-f006:**
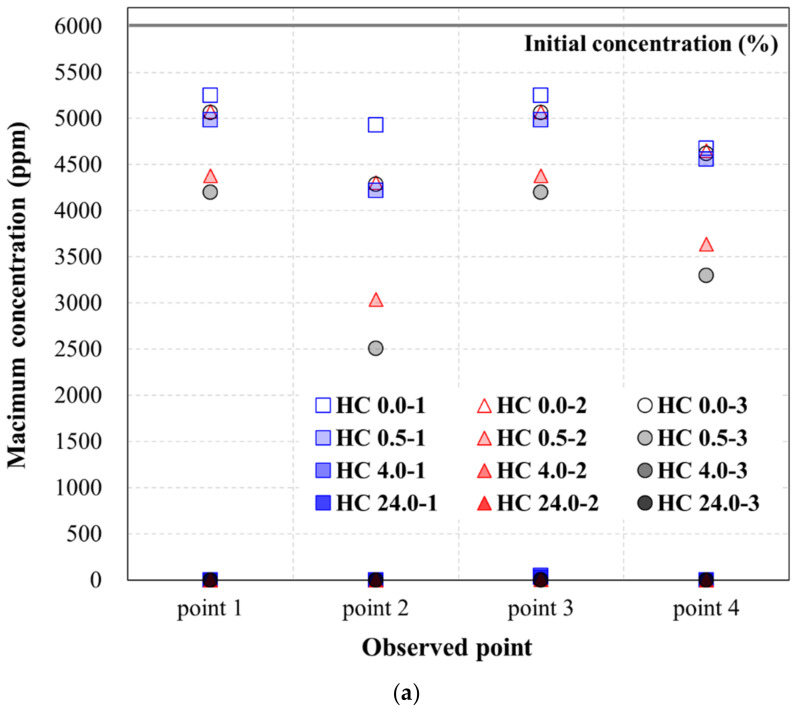
Concentration variance at the high-TPH condition: (**a**) Maximum concentration (ppm) at the observation point; (**b**) Reduction ratio of the concentration at the observation point.

**Figure 7 ijerph-19-13710-f007:**
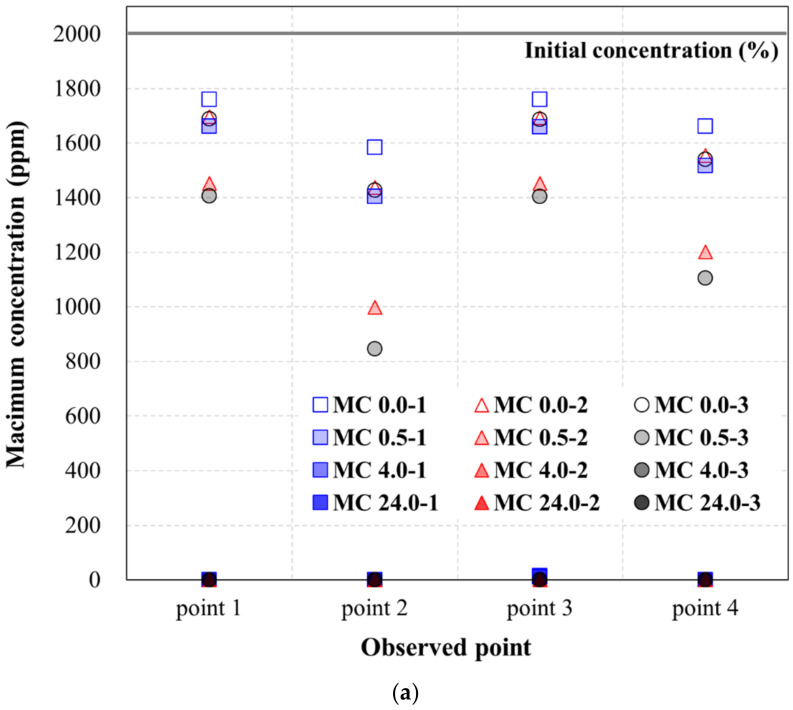
Concentration variance at the medium-TPH condition: (**a**) Maximum concentration (ppm) at the observation point; (**b**) Reduction ratio of the concentration at the observation point.

**Figure 8 ijerph-19-13710-f008:**
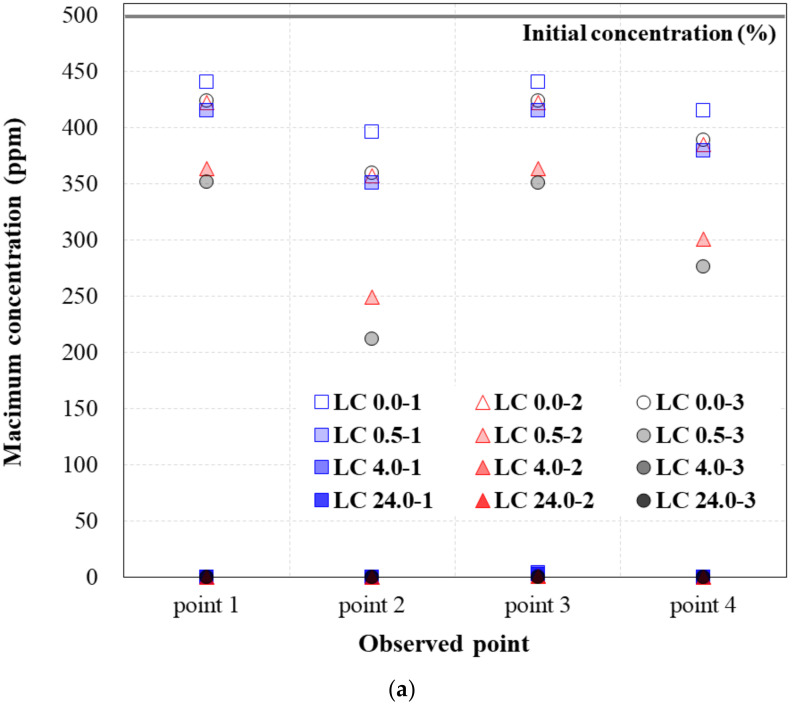
Concentration variance at the low-TPH condition: (**a**) Maximum concentration (ppm) at the observation point; (**b**) Reduction ratio of the concentration at the observation point.

**Figure 9 ijerph-19-13710-f009:**
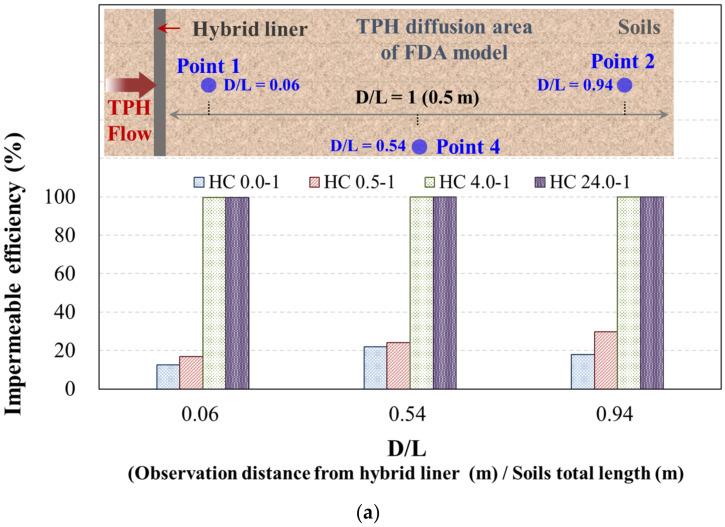
Evaluation of the impermeable efficiency using D/L: (**a**) High-TPH condition (6000 ppm); (**b**) Medium-TPH condition (2000 ppm); (**c**) Low-TPH condition (500 ppm).

**Table 1 ijerph-19-13710-t001:** Test conditions.

Classification	Reaction Time of the Hybrid Liner and TPH (H)	Specimen Length (L, cm)	Specimen Area (A, cm^2^)	Total Head (Δh, cm)
case 1	0	0.5	78.5	15
case 2	0.5	0.7	78.5	
case 3	4	0.9	78.5	
case 4	24	1.1	78.5	
case 5	48	1.5	78.5	

**Table 2 ijerph-19-13710-t002:** FDA model conditions [32].

Classification	TPH Inlet Box	Soils
Porosity	0.9	0.25
Horizontal permeability coefficient (cm/s)	1	1.0 × 10^−4^
Vertical permeability coefficient (cm/s)	1	1.0 × 10^−4^
Specific storativity (m^−1^)	10^−5^	1.0 × 10^−5^
Specific yield	0.9	0.15
Flow time of TPH (H)	96	

**Table 3 ijerph-19-13710-t003:** FDA cases.

Classification	Initial Concentration of TPH (ppm)	Reaction Time of the TPH–Hybrid Liner (H)	Head Condition (ΔP, kPa)	Permeability of the Hybrid Liner (cm/s)
HC 0.0-1	6000	0	45	9.17 × 10^−4^
HC 0.0-2			75	7.53 × 10^−4^
HC 0.0-3			105	6.46 × 10^−4^
HC 0.5-1		0.5	45	3.33 × 10^−6^
HC 0.5-2			75	2.17 × 10^−6^
HC 0.5-3			105	2.06 × 10^−6^
HC 4.0-1		4	45	2.69 × 10^−8^
HC 4.0-2			75	1.78 × 10^−8^
HC 4.0-3			105	1.58 × 10^−8^
HC 24.0-1		24	45	2.68 × 10^−8^
HC 24.0-2			75	1.64 × 10^−8^
HC 24.0-3			105	1.54 × 10^−8^
MC 0.0-1	2000	0	45	9.17 × 10^−4^
MC 0.0-2			75	7.53 × 10^−4^
MC 0.0-3			105	6.46 × 10^−4^
MC 0.5-1		0.5	45	3.33 × 10^−6^
MC 0.5-2			75	2.17 × 10^−6^
MC 0.5-3			105	2.06 × 10^−6^
MC 4.0-1		4	45	2.69 × 10^−8^
MC 4.0-2			75	1.78 × 10^−8^
MC 4.0-3			105	1.58 × 10^−8^
MC 24.0-1		24	45	2.68 × 10^−8^
MC 24.0-2			75	1.64 × 10^−8^
MC 24.0-3			105	1.54 × 10^−8^
LC 0.0-1	500	0	45	9.17 × 10^−4^
LC 0.0-2			75	7.53 × 10^−4^
LC 0.0-3			105	6.46 × 10^−4^
LC 0.5-1		0.5	45	3.33 × 10^−6^
LC 0.5-2			75	2.17 × 10^−6^
LC 0.5-3			105	2.06 × 10^−6^
LC 4.0-1		4	45	2.69 × 10^−8^
LC 4.0-2			75	1.78 × 10^−8^
LC 4.0-3			105	1.58 × 10^−8^
LC 24.0-1		24	45	2.68 × 10^−8^
LC 24.0-2			75	1.64 × 10^−8^
LC 24.0-3			105	1.54 × 10^−8^

Note: Classification: (concentration level), (reaction time of the TPH-hybrid liner)—(case No.).

**Table 4 ijerph-19-13710-t004:** Permeability coefficient of the hybrid liner.

Classification	Reactive Time of the Hybrid Liner and TPH (H)	Specimen Length (L, cm)	Permeability Coefficient (k, cm/s)	Mean Value of k (cm/s)
case 1	0	0.5	1.18 × 10^−3^	1.11 × 10^−3^
			1.08 × 10^−3^	
			1.07 × 10^−3^	
case 2	0.5	0.7	8.82 × 10^−5^	8.63 × 10^−5^
			8.56 × 10^−5^	
			8.51 × 10^−5^	
case 3	4	0.9	7.60 × 10^−7^	7.64 × 10^−7^
			7.66 × 10^−7^	
			7.62 × 10^−7^	
case 4	24	1.1	1.67 × 10^−8^	1.65 × 10^−8^
			1.58 × 10^−8^	
			1.70 × 10^−8^	
case 5	48	1.5	1.64 × 10^−8^	1.64 × 10^−8^
			1.61 × 10^−8^	
			1.68 × 10^−8^	

**Table 5 ijerph-19-13710-t005:** FDA results for high-TPH condition.

Classification	Maximum Concentration (ppm)				Reduction Ratio of the Concentration (%)			
	Point 1	Point 2	Point 3	Point 4	Point 1	Point 2	Point 3	Point 4
HC 0.0-1	5248.9	4930.0	5247.5	4676.3	12.5	17.8	12.5	22.1
HC 0.0-2	5078.8	4303.4	5072.3	4652.8	15.4	28.3	15.5	22.5
HC 0.0-3	5063.0	4283.7	5061.7	4620.3	15.6	28.6	15.6	23.0
HC 0.5-1	4985.8	4220.8	4984.2	4558.7	16.9	29.7	16.9	24.0
HC 0.5-2	4379.6	3038.4	4379.2	3636.1	27.0	49.4	27.0	39.4
HC 0.5-3	4200.9	2508.5	4197.7	3298.4	30.0	58.2	30.0	45.0
HC 4.0-1	1.4	0.0	46.1	0.0	100.0	100.0	99.2	100.0
HC 4.0-2	0.0	0.0	6.0	0.0	100.0	100.0	99.9	100.0
HC 4.0-3	0.0	0.0	3.2	0.0	100.0	100.0	99.9	100.0
HC 24.0-1	0.4	0.0	30.4	0.0	100.0	100.0	99.5	100.0
HC 24.0-2	0.0	0.0	5.7	0.0	100.0	100.0	99.9	100.0
HC 24.0-3	0.0	0.0	0.3	0.0	100.0	100.0	100.0	100.0

**Table 6 ijerph-19-13710-t006:** FDA results for the medium-TPH condition.

Classification	Maximum Concentration (ppm)				Reduction Ratio of the Concentration (%)			
	Point 1	Point 2	Point 3	Point 4	Point 1	Point 2	Point 3	Point 4
MC 0.0-1	1760.6	1584.2	1760.5	1660.7	12.0	20.8	12.0	17.0
MC 0.0-2	1696.2	1438.2	1693.9	1554.5	15.2	28.1	15.3	22.3
MC 0.0-3	1687.9	1427.9	1687.5	1540.7	15.6	28.6	15.6	23.0
MC 0.5-1	1660.5	1404.0	1660.1	1517.5	17.0	29.8	17.0	24.1
MC 0.5-2	1453.2	997.7	1453.0	1201.2	27.3	50.1	27.4	39.9
MC 0.5-3	1405.3	845.5	1404.3	1105.7	29.7	57.7	29.8	44.7
MC 4.0-1	0.5	0.0	15.4	0.0	100.0	100.0	99.2	100.0
MC 4.0-2	0.0	0.0	2.0	0.0	100.0	100.0	99.9	100.0
MC 4.0-3	0.0	0.0	1.0	0.0	100.0	100.0	100.0	100.0
MC 24.0-1	0.1	0.0	10.3	0.0	100.0	100.0	99.5	100.0
MC 24.0-2	0.0	0.0	1.1	0.0	100.0	100.0	99.9	100.0
MC 24.0-3	0.0	0.0	0.0	0.0	100.0	100.0	100.0	100.0

**Table 7 ijerph-19-13710-t007:** FDA results for the low-TPH condition.

Classification	Maximum Concentration (ppm)				Reduction Ratio of the Concentration (%)			
	Point 1	Point 2	Point 3	Point 4	Point 1	Point 2	Point 3	Point 4
LC 0.0-1	440.2	396.0	440.1	415.2	12.0	20.8	12.0	17.0
LC 0.0-2	422.0	357.0	421.9	385.2	15.6	28.6	15.6	23.0
LC 0.0-3	424.1	359.5	423.5	388.6	15.2	28.1	15.3	22.3
LC 0.5-1	415.1	351.0	415.0	379.4	17.0	29.8	17.0	24.1
LC 0.5-2	363.3	249.4	363.3	300.3	27.3	50.1	27.4	39.9
LC 0.5-3	351.3	211.4	351.1	276.4	29.7	57.7	29.8	44.7
LC 4.0-1	0.1	0.0	3.8	0.0	100.0	100.0	99.2	100.0
LC 4.0-2	0.0	0.0	0.5	0.0	100.0	100.0	99.9	100.0
LC 4.0-3	0.0	0.0	0.2	0.0	100.0	100.0	100.0	100.0
LC 24.0-1	0.0	0.0	2.5	0.0	100.0	100.0	99.5	100.0
LC 24.0-2	0.0	0.0	0.3	0.0	100.0	100.0	99.9	100.0
LC 24.0-3	0.0	0.0	0.0	0.0	100.0	100.0	100.0	100.0

## Data Availability

Not applicable.

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
