# Peer review of "Simulation on the Permeability Evaluation of a Hybrid Liner for the Prevention of Contaminant Diffusion in Soils Contaminated with Total Petroleum Hydrocarbon"

_ijerph, 2022, doi:10.3390/ijerph192013710_

Round 1

Reviewer 1 Report

This paper, entitled Simulation on permeability evaluation of hybrid liner for prevention of contaminant diffusion in soils contaminated with total petroleum hydrocarbon, is a scholarly work and can increase knowledge on this domain. The authors provide an interesting and original study, the content is relevant to IJERPH. The abstract and keywords are meaningful. The manuscript is quite well written and well related to existing literature.

I have some general and specific comments:

- about Figure 2, the pictures on the bottom are too small and should be resized for a better understanding and visibility. The scheme of Figure 1should be enlarged too.

- How were selected or screened the materials for the building of the hybrid liner? Please provide more details about synthesis of ths hybrid liner. What are the costs of synthesis? What about degradability and lifetime of such material?

- What is the maximal size or area, thickness of this material? I understand that such approach was carried out at labscale. What about the transfer at highest scale of such approach? What is the feasability to produce larger piece of hybrid liner and to implement on real location? What is the real applicability of such proposal?

- About length and area, I don't understand why aera was 78.5 cm2 when length increased?  (table 1). What is the accuracy of the dimension?

- Please provide error bars in Figure 5.

- Please provide accuracy of data in Table 5, Table 6and in Table 7.

- Figures 6 and 7 are too small, same comment for Figure 8 and Figure 9.

- Is there any control or comparison with other existing material with similar or closed characteristics for this application? By this way, the authors should provide more details and discussion about the novelty and originality of their work and this approach. This point should be discussed better in the manuscript.

- What are the perspectives of this work? What are the future experiments?

As it, this manuscript is not fully acceptable for publication and requires some amendments and additional data or information. I recommend to revise this manuscript according to the previous comments.

Reviewer 2 Report

The manuscript "Simulation on Permeability Evaluation of Hybrid Liner for Prevention of Contaminant Diffusion in Soils Contaminated with Total Petroleum Hydrocarbon" submitted in International Journal of Environmental Research and Public Health. It's a nice research with collaboration of Korean researchers about simulation of pollutants in contaminated soils. The topic is so nice, but there are some major concerns in this MS that I can't accept. Finally, I present some suggestions for improving the quality of this MS as following:

1.       In the abstract, the authors have more spoken about the necessity of the research, and there is not a quantitative report about results. It is strong recommended to complete and revise it. Finally, talk briefly about the conclusion of the research.

2.      There are repeated words between the title and keywords, please revise them, such as: hybrid liner; permeability; contaminant; diffusion.

3.      It is recommended to talk more about the title of the article and environmental problems in a paragraph in the introduction.

4.      The materials and methods well written, congratulations.

5.      One of the significant concerns is that the authors should carefully develop a discussion section to talk about the significance, shortages or advantages of the methods you proposed, the reliability and meaning of your results (compared to other related studies) etc.

6.      I didn’t find anything related to the limitations of the research. It is recommended to complete it.

7.       Please be sure that all the references cited in the manuscript are also included in the reference list and vice versa with matching spellings and dates.

8.      Finally, I checked plagiarism detection of this research and the similarity is 24% and there some concerns, please checked attached file.  

Reviewer 3 Report

Dear Editor:

Thank you for giving me the opportunity to revise the manuscript that was submitted to “Computers and Geosciences”. In general, the manuscript contains an interesting topic; but it required several modification. In this regard, the following comments are requested to be addressed by the authors:

Comment 1: The English of the paper is readable; however, I would suggest the authors to have it checked preferably by a native English-speaking person to avoid any mistakes.

Comment 2: The necessity & novelty of the manuscript should be presented and stressed in the “Introduction” section.

Comment 3: Provide a literature of the methods developed/applied on permeability in “Introduction”. The use of a table to demonstrate the advantage-disadvantage of these methods can be useful. Towards the end, mention the superiority & repeat the novelty of your work.

Comment 4: Please add a subsection clearly articulating the main limitations, wider applicability of your methods, and findings in the “Discussion” section.

Comment 5: The authors should deepen the discussion.

Comment 6: I would suggest that the authors review and include the following studies to improve the manuscript.

1. Nikbakht, M., Sarand, F. B., Esmatkhah Irani, A., Hajialilue Bonab, M., Azarafza, M., & Derakhshani, R. (2022). An Experimental Study for Swelling Effect on Repairing of Cracks in Fine-Grained Clayey Soils. Applied Sciences, 12(17), 8596.

2. Song, X. F., Wei, J. F., & He, T. S. (2009). A method to repair concrete leakage through cracks by synthesizing super-absorbent resin in situ. Construction and Building Materials, 23(1), 386-391.

Best regards,

Round 2

Reviewer 3 Report

Accepted.